# Inter-Domain Fusion for Enhanced Intrusion Detection in Power Systems: An Evidence Theoretic and Meta-Heuristic Approach

**DOI:** 10.3390/s22062100

**Published:** 2022-03-09

**Authors:** Abhijeet Sahu, Katherine Davis

**Affiliations:** Electrical Engineering Department, Texas A&M University, College Station, TX 77843, USA; katedavis@tamu.edu

**Keywords:** Dempster–Shafer theory, intrusion detection system, genetic algorithm

## Abstract

False alerts due to misconfigured or compromised intrusion detection systems (IDS) in industrial control system (ICS) networks can lead to severe economic and operational damage. However, research using deep learning to reduce false alerts often requires the physical and cyber sensor data to be trustworthy. Implicit trust is a major problem for artificial intelligence or machine learning (AI/ML) in cyber-physical system (CPS) security, because when these solutions are most urgently needed is also when they are most at risk (e.g., during an attack). To address this, the Inter-Domain Evidence theoretic Approach for Inference (IDEA-I) is proposed that reframes the detection problem as how to make good decisions given uncertainty. Specifically, an evidence theoretic approach leveraging Dempster–Shafer (DS) combination rules and their variants is proposed for reducing false alerts. A multi-hypothesis mass function model is designed that leverages probability scores obtained from supervised-learning classifiers. Using this model, a location-cum-domain-based fusion framework is proposed to evaluate the detector’s performance using disjunctive, conjunctive, and cautious conjunctive rules. The approach is demonstrated in a cyber-physical power system testbed, and the classifiers are trained with datasets from Man-In-The-Middle attack emulation in a large-scale synthetic electric grid. For evaluating the performance, we consider plausibility, belief, pignistic, and general Bayesian theorem-based metrics as decision functions. To improve the performance, a multi-objective-based genetic algorithm is proposed for feature selection considering the decision metrics as the fitness function. Finally, we present a software application to evaluate the DS fusion approaches with different parameters and architectures.

## 1. Introduction

The increase of advanced control and communication technologies within an electric power grid can also increase its vulnerability to cyber intrusions. Several industrial control systems (ICS)-targeted attacks such as Stuxnet [1], Ukraine [2], and Mumbai are well known for their advanced concept of operations and physical impacts. The criticality of power grid infrastructure necessitates the design of resilient detection and defense mechanisms against such attacks.

The challenge is that detection is subject to the stochastic and uncertain nature of attacks. Intrusion Detection Systems (IDS) commonly rely on rule-based policies (signature-based) or deviations from a baseline (behavioral-based) to detect cyber intrusions. These systems produce false alarms, both false negatives and false positives. Signature-based IDSs result in higher false negatives for stealthy and zero-day attacks. Behavioral-based IDSs often result in high false positives. High false positives are detrimental to an organization’s effective threat response because they cost time and money for security professionals to investigate, and they erode an organization’s trust in the system’s results. False negatives also pose a significant threat, since an undetected attack may escalate privileges to result in increased damage or loss to the organization’s assets.

ICS are monitored and controlled using supervisory control and data acquisition (SCADA) and other operational technology (OT) networks. These networks are highly static, as they are designed to meet a number of industry based criteria, e.g., the North American Electric Reliability Corporation (NERC) reliability standards [3], including Critical Infrastructure Protection (CIP) that focuses on grid cyber and physical security. For ICS, intrusion detection may be customized based on process-data analysis, control-command analysis, and with help of an ICS physical model [4]. While security tools such as IDSs and firewalls provide key functions, they are typically assumed to be trustworthy. Furthermore, obtaining the data needed for theoretical models to improve the function of such security tools is a challenge, as even behavioral-based IDSs do not have enough intrusion information to build the statistical models [5]. The lack of trust in the IDS creates uncertainty in the evidences from sensors.

To address the lack of trust of the IDS sensors, various hardware and software-based authentication schemes have been proposed in the literature from different research groups. For instance, a Physical Clonable Functions (PUF)-based PUF-Cash Multiple Trusted Third Party (TTP) protocol is proposed for artificial intelligence (AI)-based Electronic Money [6], which leverages AI for the selection of an optimal subset of slave TTP for communication from the Master TTP for secure transaction. An extended Diffie–Hellman key exchange algorithm as a lightweight and secure authentication scheme based on MQTT subscribe and publish protocol is proposed for distributed fog computing in [7]. These software-based solutions address the issue of higher false alarms through changing the authentication protocol for the sensors. Unlike their approaches, our proposed solution addresses the uncertainty in alerts by utilizing the features from the headers and payloads from the existing protocols.

To address these challenges, this work presents a cyber-physical power system intrusion detection system based on the theory of uncertainty, called the **I**nter-**D**omain **E**vidence theoretic **A**pproach for **I**nference in cyber-physical power systems (IDEA-I). It addresses the problem of high false alarms in IDS through the developed solution that leverages the fusion of evidence by domain and location using Dempster–Shafer (DS) rules of combination. IDEA-I is based on an *autonomous* data fusion architecture [8], where the features extracted are fed to the classifiers or estimators for decision making before they are fused. This is *decision-level* fusion, where each sensor performs individual processing to produce an estimate, and then these estimates are combined in the fusion process. Numerous methods are possible in fusion, such as voting methods, Bayesian inference, DS methods, and generalized evidence processing theory [8]. DS and Bayesian inference are appropriate to the autonomous fusion architecture [9], as these fusion algorithms are fed with the probability distributions computed from the classifiers or the estimators.

In IDEA-I, we propose the usage of Dempster–Shafer Theory of Evidence (DSTE) for network detection in power system control networks. This approach provides value in how it handles uncertainty due to its ability to quantify unknowns. Specifically, two advantages are (1) its ability to deal with the lack of prior probabilities for various events and (2) its ability to combine evidences from multiple sources [10].

The major contributions of this paper are as follows:1.A cyber-physical power system intrusion detection system IDEA-I is proposed that improves intrusion detection by inferring cyber-physical state information to improve situational awareness based on the fundamentals of DSTE, various rules of fusion, and decision criteria.2.A method for computing mass functions for stochastic cyber-physical parameters, from the detection probability computed in our prior work on data fusion [11], is proposed and evaluated in IDEA-I. The performance based on two different architectures, location and location-cum-domain based fusion, using IDEA-I is evaluated.3.IDEA-I is extended to formulate a feature selection unconstrained optimization problem and solved using the Non-dominating Sorted Genetic Algorithm (NSGA) [12] to improve IDEA-I accuracy.4.IDEA-I is developed as a software tool that includes the development of a DSTE library in C#. The application is used to evaluate the performance of the proposed fusion algorithm for varying scenarios and parameters.

The paper is organized as follows. Section 2 develops the IDEA-I framework based on DSTE and describes how the method would need to work in cyber-physical power systems. A genetic algorithm (GA)-based optimization problem is proposed in Section 3 for feature selection to improve the IDEA-I performance. Section 4 presents the experimental setup in our cyber-physical power system testbed, the use cases that were designed to test, IDEA-I and their implementation. Then, Section 4.3 introduces the two types of architecture proposed for the fusion. We compare the approach and results with the centralized-based fusion and other decision-level fusions such as Bayesian inference. The overall results are discussed in Section 5, and Section 7 concludes the paper.

## 2. Development of IDEA-I from Dempster–Shafer Theory and Combination Rules

Uncertainty can be classified into aleatory or epistemic uncertainty [13]. *Aleatory* uncertainty is caused by random behavior of the system, while *epistemic* uncertainty is caused by lack of knowledge of the system. Under normal operation of an ICS network, most traffic is not random; hence, aleatory uncertainty is not significant. Under a compromise, the system state is ignorant rather than stochastic; hence, the uncertainty in events is epistemic. For example, a zero-day attack cannot be detected by knowledge-based or signature-based IDS due to a lack of information about the intrusions. Dempster and Shafer introduced the belief function for modeling epistemic uncertainty for reasoning under uncertainty. Quantifying uncertainty with a precise measure is difficult, and hence, a measure of probability as an interval is considered. Three major frameworks for interval-based representation of uncertainty are the following: (a) imprecise probability, (b) possibility theory, and  (c) Demspter–Shafer Theory of Evidence (DSTE). DSTE is preferred because of its high degree of theoretical development, better relationship with traditional probability theory, large engineering applications in the past few years, and the versatility of the theory to represent and combine different types of evidences.

In evidence theory, logs at each sensor act as evidence that are considered for reasoning about an event. Theoretically, there are four types of evidence: (a) consonant, (b) consistent, (c) arbitrary, and (d) disjoint [14]. The data of a control network, with an IDS for cyber intrusion detection and a bad data detector for power system state estimation, can be collectively considered as disjoint evidence due to their different purposes of deployment. Traditional probability theory cannot handle consonant, consistent, or arbitrary evidence without resorting to assumptions in distributions. DSTE can handle all these kinds by combining a notion of probability with the traditional conception of sets.

Research on DSTE includes applications for unsupervised classification in remote sensing, computing basic belief assessment (BBA) and rules of combination in hidden layers of a neural network [15], wireless networks [16], and autonomous mobile robots. DSTE is also being applied to network security [17]. In [18], multi-source alarm information is fused through DSTE, which is associated with nodes’ vulnerability information, integrated with the severity of threats for situational assessment of network security. A network anomaly detector with enhanced reliability with low false alarms is proposed using DSTE [19]. An IDS is proposed in [17] where the mass function is computed based on the incoming and outgoing traffic ratio, service rate, and the prior knowledge in the domain of DDoS attacks. A distributive and collaborative-based IDS is proposed using DSTE for fusion data from multiple nodes [20], where the detection is done collaboratively and the decision is distributed among all nodes. The presented work, IDEA-I, is the first to leverage DS theory for the purpose of classification based on the dataset [11] generated from Man-in-The-Middle (MiTM) attacks in a cyber-physical power system testbed [21].

DSTE suffers from major drawbacks of its computational requirements and the challenges it encounters while eliciting the probability masses from multiple evidence [13]. Hence, to address this, we have proposed the use of formulating an optimization problem by taking the decision function from DSTE as the objective function for feature selection to train the classifier. Since there are multiple decision metrics, we employ a multi-objective optimization problem and solve using a meta-heuristic GA approach. GA has been used extensively in network intrusion detection such as flow-based traffic characterization [22], feature selection [23], etc. GA in the DSTE framework was proposed in the turbine maintenance optimization problem [13]. In this work, we present GA with DSTE in cyber-physical security for feature selection.

### 2.1. IDEA-I Framework

The IDEA-I framework is illustrated in Figure 1. The *Datasets* are sensor data extracted from substation networking devices, DNP3 Master, and Outstations from our testbed [21]. DNP3 [24] is a protocol used in SCADA systems for monitoring and controlling field devices. *Data Pre-Processing* is performed before training the ML-based *IDS Classifier*. The output from the classifier from different sensors carries data of varying timestamps, which synchronized with the *Mean Value-Based Time Synchronization* block. The calculations for *Mass Function Computation*, *DS Rules of Combination*, and the *Decision Function* blocks of DSTE are detailed in this section. The decision function is used in the fitness function in the version 2 of NSGA i.e., *NSGA-2 Based Feature Selection* block (Section 3) to again filter the features in the Data Pre-Processing block.

Data fusion in a cyber-physical system utilizes data from the physical and cyber sensors as evidence to generate belief functions for the hypothesis, e.g., “root vulnerability exploitation”. Cyber-physical frameworks [25,26] have been proposed that identify critical assets and contingencies using power system simulators, graph theories, and dynamic programming. For example, ref. [25] builds a partially observable Markov Decision Process (POMDP) model of the grid network and all possible attack paths. The robustness of such a framework depends on fusion of information such as network access policies, firewall rulesets, physical sensors, etc. The accuracy of the transition probabilities of the security states in the POMDP model depend on the amount of data accrued in real time. However, uncertainty is present due to the unavailability of a full view of the adversary’s steps and monitoring limitations. The presented IDEA-I addresses this gap through its use of DSTE for power system cyber-physical situational awareness.

### 2.2. Dempster–Shafer Theory of Evidence (DSTE)

DSTE is separated from the basic probability theory on the basis of the manner one distributes the probability density or mass based on the type of random variables. For example, probability theory assigns 0.5 to both Head and Tail for the toss of an unbiased coin. However, DS theory assigns a 0 belief to Head and Tail but assigns a 1 belief to the set {Head,Tail}, i.e., *“Either Head or Tail”*. DS does not compel picking a probability when there is no evidence. This approach provides three kinds of answers: Yes, No, and *Don’t Know*. Allowing the third option, i.e., *ignorance*, can make evidential reasoning valuable when there are not enough data to validate a hypothesis.

DSTE is concerned with *bounds* for probabilities of provability rather than computing probabilities of truth. The two bounds are called *belief* and *plausibility*. Equivalent to the state space in probability, there is a set of mutually exclusive and exhaustive hypotheses denoted by Ω, which is also called the *Frame of Discernment*. The set of all possible subsets of Ω, including itself and the null set *O*, is called a power set and designated by $2^{\Omega }$. Thus, the power set comprises all possible hypotheses or so-called *focal elements*.

### 2.3. Basic Belief Assignment in DSTE

The basic belief assignment (BBA) function or the mass distribution function (*m*) distributes the belief over the power set of the frame of discernment. Subsets *A* of Ω such that m(A)>0 are called *focal sets* of *m*. The mass distribution functions are computed by utilizing the detection probability scores from the ML-based IDS classifier and performing Mean Value-Based Time Synchronization, which is explained further. Then, the functions considered in the Decision Function block (Figure 1) are used for validating a hypothesis, as presented in Section 2.8.

### 2.4. Supervised Learning Based IDS

Different Supervised Learning (SL)-based classifiers are used in the Data Fusion Engine [11]. The probability scores based on the classifier’s output for each data point are considered for computing the mass function from each evidence. In the testbed, the IDSs are trained at three different locations in the network: (a) DNP3 outstation, (b) DNP3 master, and the (c) Substation Router, which are thoroughly presented in Section 4.1. Seven types of SL-based IDS classifiers are trained: (a) Support Vector Classifier (SVC), (b) K-Nearest Neighbor (kNN), (c) Decision Tree (DT), (d) Random Forest (RF), (e) Gaussian Naive Bayes (GNB), (f) Bernoulli Naive Bayes (BNB), and (g) Multi-layer Perceptron (MLP) to compute the probability scores for different use cases with varying poll rates and the polled number of outstations.

### 2.5. Mean Value-Based Time Synchronization

The classifier’s probability scores of detection and the time stamp may vary for different locations of the IDS. The sample times will also vary. Hence, a time resolution window res is selected to compute the average of the probability scores from samples existing in that window and store the average probability score. This ensures time synchronization for fusion by location. The lower the window size, the higher the time resolution will be, but more noise will be present in the decision function. The impact of resolution is studied considering accuracy of the fusion technique for res = 5, 10, 15, 20 s.

### 2.6. Mass Function Computations

The mass distribution is computed based on the probability score. The frame of discernment for the given IDS problem is given by {attack};{no_attack};{attack,no_attack};{∅}. If the probability of intrusion for a data point *t* is say *a*, then the dogmatic belief mass distribution is set to be the following,
(1)$$\begin{array}{c} m_{t}\left(\emptyset \right)=-1\sum_{i=1}^{2}{\left(m_{t}\left(i\right)-1/2\right)}^{2}\\ m_{t}\left(attack\right)=a*\left(1-m_{t}\left(\emptyset \right)\right)\\ m_{t}\left(no\_attack\right)=\left(1-a\right)*\left(1-m_{t}\left(\emptyset \right)\right) \end{array}$$
where the first mass belief distribution $m_{t}\left(\emptyset \right)$ quantifies uncertainty, as per the variance of the probability scores considered in [27]. However, since most if not all states of belief are based on imperfect and not entirely conclusive evidence, non-dogmatic belief functions should be considered where m(Ω) is very small [28], say ϵ and not zero. In this scenario, a different belief mass distribution is proposed: (2)$$\begin{array}{c} m_{t}\left(\emptyset \right)=-1\sum_{i=1}^{2}{\left(m_{t}\left(i\right)-\left(1-\epsilon \right)/2\right)}^{2}\\ m_{t}\left(attack\right)=a*\left(1-m_{t}\left(\emptyset \right)\right)*\left(1-\epsilon \right)\\ m_{t}\left(no\_attack\right)=\left(1-a\right)*\left(1-m_{t}\left(\emptyset \right)\right)*\left(1-\epsilon \right)\\ m_{t}\left(attack,no\_attack\right)=m\left(\Omega \right)=\epsilon *\left(1-m_{t}\left(\emptyset \right)\right) \end{array}$$

### 2.7. Rules of Combination

The purpose of aggregation of information is to summarize a collection of data, whether the data are coming from a single source or from multiple sources.

#### 2.7.1. Dempster’s Rules of Combination (DRC)

Dempster’s rules of combination is a procedure for combining independent pieces of evidence. The requirement of establishing the independence of sources is an important philosophical question. From a set theoretic standpoint, these rules can potentially occupy a continuum between *conjunction* (AND-based on set intersection) and *disjunction* (OR-based on set union) [29]. Where all evidence are reliable, a *conjunctive* operation is appropriate, while for one reliable source, *disjunctive* operation is preferred. Hence, in the domain of intrusion detection, one should prefer the disjunctive rule. The normalized conjunctive fusion rule of combination is given by:(3)$$\begin{array}{c} m_{1,2}\left(A\right)=\left(m_{1}⊕m_{2}\right)\left(A\right)\\ =\frac{1}{1-K}\sum_{B\cap CA\neq }m_{1}\left(B\right)*m_{2}\left(C\right) \end{array}$$
where
(4)$$K=\sum_{B\cap Ceq}m_{1}\left(B\right)*m_{2}\left(C\right)$$

The disjunctive rule of combination is given by:(5)$$m_{1,2}\left(A\right)=\left\{\begin{array}{cc} \sum_{A=B\cup C} & m_{1}\left(B\right)m_{2}\left(C\right),\;A\neq \emptyset \\ 0, & \;\;\;\;\;\;\;\;A=\emptyset \end{array}\right.$$

#### 2.7.2. Combine Cautious (CC)

The Combine Cautious rules of combination are based on the work [28]. Conventional DS rules of combination require the evidence from multiple sources to be distinct or independent, which may not be true in reality. Many works have developed mechanisms to overcome this limitation, but they were limited to at most two focal sets. DRC methods were extended to separable belief functions, but since all belief functions are not separable, the conventional method was not further extended. The operators in DRC need to be *associative*, *commutative*, and *idempotency*. Many developed rules of combination either did not obey those requirements or were not scalable for large focal sets. Moreover, the conjunctive rule assumes that the belief functions to be combined are induced from reliable sources of information. Due to these challenges in DRC, the CC method of combination is also considered.

DSTE manages imperfect information through two levels: *credal* and *pignistic*. *Pignistic* is related to a probability that a rational person will assign to an option during decision making, while *credal* defines the probability based on belief. In the DRC, within the *credal level*, the evidences are quantified and aggregated, and the decisions rules are implemented at the *pignistic level* based on Equation (Equation 10) [30]. While in the CC, the commonality function *q* and the weight function are computed:(6)$$\begin{array}{c} q\left(B\right)=\sum_{j:B\cap A_{j}}m\left(A_{j}\right) \end{array}$$
(7)$$\begin{array}{cc} w(A) & =\prod_{B⊇A}q{\left(B\right)}^{{\left(-1\right)}^{|B|-|A|+1}}\\ \; & =\left\{\begin{array}{cc} \frac{\prod_{B⊇A,|B|\notin 2N}q\left(B\right)}{\prod_{B⊇A,|B|\in 2N}q\left(B\right)} & \;\mathrm{if}\;|A|\in 2N\\ \frac{\prod_{B⊇A,|B|\in 2N}q\left(B\right)}{\prod_{B⊇A,|B|\notin 2N}q\left(B\right)} & \;\mathrm{otherwise}\; \end{array}\right. \end{array}$$
where 2N denotes the set of even natural numbers.

### 2.8. Decision Criteria

#### 2.8.1. Belief and Plausibility Score

After the fused mass function are computed using disjunctive, conjunctive, and cautious combine rules, the belief and plausibility scores are calculated as:(8)$$\begin{array}{c} bel\left(B\right)=\sum_{j:A_{j}\subset B}m\left(A_{j}\right) \end{array}$$
(9)$$\begin{array}{c} pls\left(B\right)=\sum_{j:A_{j}\cap B\neq }m\left(A_{j}\right) \end{array}$$

The belief in hypothesis *B* is the sum of masses of elements that are subsets of *A*, $A_{j}$. Plausibility is the sum of all the mass of the sets $A_{j}$ that intersect with the set *B*.

#### 2.8.2. Pignistic Score

To make a rational decision, we propose to transform beliefs into pignistic probability functions through the generalized pignistic transformation (GPT) [30]. The pignistic transformation is based on the following equation,
(10)$$P\left\{A\right\}=\sum_{X\in 2^{\Theta }}\frac{\left|X\cap A\right|}{\left|X\right|}m\left(X\right)$$
where |A| is the number of the worlds present in the set *A*, and *X* are the other components in the frame of discernment. Usually, decisions are made by computing the expectation over multiple simulations, using the pignistic P{.} as the probability function needed to compute expectations. Conventionally, one uses the maximum of the pignistic probability as decision criterion.

#### 2.8.3. General Bayesian Theorem (GBT)

In the literature, DS theory has two main views: DS theory of evidence (DSTE) and DS theory of generalization of probability. The second view handles a wider variety of data imperfection as well as allows one to perform Bayesian Inference within a DS theoretic framework [31]. Considered in the second view, the GBT is a generalization of Bayes’ theorem, except that the conditional probabilities in Equation (Equation 11) are replaced by belief functions, and the a priori belief function on Θ is vacuous.
(11)$$P\left(\theta_{i}|x\right)=\frac{P\left(x|\theta_{i}\right)P_{0}\left(\theta_{i}\right)}{\sum_{j}P\left(x|\theta_{j}\right)P_{0}\left(\theta_{j}\right)}\;\forall \theta_{i}\in \Theta$$

## 3. Genetic Algorithm for Feature Selection

Feature selection is difficult for intrusion detection due to uncertainty, which increases due to CPS complexity. Leveraging feature reduction techniques such as PCA can assist in improving detection accuracy, but feature transformation can obfuscate the meaning of features. Hence, IDEA-I adopts non-transformable techniques for feature reduction using optimization techniques while also considering system uncertainty.

The feature selection problem in a stochastic system relies on the epistemically uncertain parameters. This problem is formulated as a multi-objective optimization problem with uncertain fitness functions (i.e., the belief, plausibility, and pignistic scores). In this context, the objective of the present work is to propose a feature selection technique by propagating the uncertainties of the conventional classifiers onto the fitness values and formulating the solution of the GA as a binary encoding. A GA-based meta-heuristic approach is adopted for feature selection, and the initial population consists of chromosomes of randomly selected features. The fitness functions are obtained for the different evidences, which were further used for selection, mutation, and cross-over operation. Since there are multiple decision criteria in this problem, a multi-objective problem is formulated.

### 3.1. Non-Dominated Sorting Genetic Algorithm ver. 2 (NSGA-2)

A single fitness function cannot provide an optimal solution for the multiple decision metrics considered in the DSTE framework. Hence, multi-objective GA algorithms need to be explored. NSGA [12] has been found to solve multi-objective problems efficiently. In this paper, a faster version of NSGA (NSGA-2) is used to solve the feature selection problem.

The algorithm for NSGA-2 is given in Algorithm 1. It involves two steps: (a) From population, $P_{t}$, at iteration *t*, offspring solution, $Q_{t}$, is obtained using the selection, mutation, and crossover operations (Lines 12–15). First, the union of $P_{t}$ and $Q_{t}$, non-dominated sorting is performed to obtain solutions at different Pareto-front levels (Lines 2–3). (b) In the second step, while the next population set $P_{t+1}$ is obtained by sequentially adding the elements in the obtained Pareto fronts, starting with 1 until the condition $|P_{t+1}\left|+\right|F_{i}|\leq N$ is satisfied (where $F_{i}$ is the solution in the $i^{th}$ front, and *N* is the maximum size of the population), for the selection of the elements in $F_{i}$, crowding-distance computation using the fitness function in each front (Line 6) is performed to obtain diverse solutions (Lines 5–9).
**Algorithm 1** Algorithm of NSGA-21:**while** termination criteria **do**2:    $R_{t}\leftarrow P_{t}\cup Q_{t}$3:    F← non_dominated_sorting($R_{t}$)4:    $P_{t+1}\leftarrow \phi ;i\leftarrow 1$5:    **while** $|P_{t+1}\left|+\right|F_{i}|\leq N$ **do**6:        $C_{i}\leftarrow$ crowd_sourcing_assignment($F_{i}$)7:        $P_{t+1}\leftarrow P_{t}\cup F_{i}$8:        i=i+19:    **end while**10:    $F_{i}\leftarrow sort\left(F_{i},C_{i},desc\right)$11:    $P_{t+1}\leftarrow P_{t+1}\cup F_{i}\left[1:(N-\right|P_{t+1}\left|)\right]$12:    $Q_{t+1}\leftarrow selection\left(P_{t+1},N\right)$13:    $Q_{t+1}\leftarrow mutation\left(Q_{t+1}\right)$14:    $Q_{t+1}\leftarrow crossover\left(Q_{t+1}\right)$15:    t←t+116:**end while**

### 3.2. Problem Formulation

The objective of the feature reduction problem is to minimize the error with reference to attack labels A(t), obtained from Snort IDS, over the sampled time throughout the simulation, so as to identify the least number of features that need to be considered for training the classifiers,
(12)$$\min\;F_{k}=\sum_{t=0}^{N_{sim}}\left|f_{k}\left(t\right)-A\left(t\right)\right|$$
where k∈K, *K* is the set of all decision metrics such as fused belief, plausibility, pignistic, and GBT functions, $f_{k}$ is their corresponding scores after the fusion operations, as presented in Section 2 and Section 2.8, respectively, and $N_{sim}$ is the simulation duration. The decision variables are binary encoded, indicating whether a feature is selected for training the classifier or not. The $f_{k}\left(t\right)$ at time *t* depends on the feature selected for training. A(t)∈0,1 depending on the attack window, i.e., A(t)=1 during attack or else A(t)=0.

### 3.3. Analysis of Computation Time of IDEA-I

IDEA-I collectively involves the DRC along with the NSGA-2 algorithm for feature selection. The time complexity of the DRC depends on two parameters: the size of the frame of discernment as well as the number of mass functions being combined [32]. The size of the frame of discernment grows exponentially with the number of variables (or class in the case of evidence from classifiers). Since we have considered two classes (alert and no alert) for the SL-based classifier, this factor does not impact the computation. Meanwhile, the amount of evidence that is fused will depend on the fusion architecture. For the meta-heuristic algorithm for feature selection, the parameters such as (a) population size for each generation, (b) cross-over points in mutation, and (c) offspring size in every generation can affect the computation time. The time complexity of the NSGA-2 algorithm is $O(MN^{2})$ [12], where *M* is the number of objective functions, and *N* is the population size.

## 4. Testbed and Fusion Architecture

### 4.1. Testbed Architecture

Before discussing the fusion architecture, we present the testbed that is producing the data during the emulation of different MiTM attacks. The testbed emulates cyber-physical power systems using Common Open Research Emulator (CORE) and Power World Dynamic Studio (PWDS), with SCADA clients and servers of DNP3 (an OpenDNP3 master and a RTAC-based master), Snort IDS, and data storage, fusion and visualization software (Elasticsearch, Logstash, and Kibana (ELK) stack), as shown in Figure 2. The evidences are collected at three locations: DNP3 Master, Substation Router, and server hosting PWDS. Details on the testbed’s architecture, use cases, and fusion are published in [11,21,33], respectively.

### 4.2. Threat Model: Modifying Measurements and Commands

The objective of the intruder is to disrupt grid operations through False Command and Data Injection, whose impacts are detailed in [21,33]. Four use cases with this threat model are considered here. The first two are pure FCI, while the next two are a mix of FDI and FCI, as follows. *UC 1*: Selected branches’ binary direct operate command are changed from a *CLOSE* to a *TRIP* command. *UC 2*: Selected generators’ set-point command are modified. *UC 3*: The readings of generator outputs are modified in the DNP3 read response traffic, forcing the operator to raise the set-point, further modifying the generation set-point to a low value. *UC 4*: The adversary first follows *UC 3*; then, it modifies the read response packet of the preceding packets based on the actual set point, making the master unaware of the contingency created.

### 4.3. Fusion Architectures

Two types of fusion architectures are proposed and adopted, as shown in Figure 3.

#### 4.3.1. Fusion by Location (FL)

In the first case, we explore the performance through fusion by location based on the probability scores obtained from classifiers trained with both cyber and physical features at the substation router, DNP3 master, and outstation.

#### 4.3.2. Fusion by Location and Domain (FLD)

In the second case, we fuse by location as well as domain, utilizing the probability scores obtained from classifiers trained with pure cyber and physical features.

### 4.4. IDEA-I as a Software-Based Solution

IDEA-I is considered to be a software-based solution, as it incorporates Application Programming Interfaces (APIs) and the protocol parsers to extract features from a power system simulation server, PWDS, and packet header and payload information from real-time network traffic using Packetbeat, Pyshark, etc., to further train the SL-based classifiers, considering alerts from IDSes. Furthermore, it uses a framework for fusing the alerts from multiple locations in the emulated network using the DSTE. Unlike various hardware-based approaches, a software-based approach is adopted in the current approach of intrusion detection, because the physical sensors are emulated within PWDS, and the cyber sensors such as Snort, Packetbeat, and pyshark are software-based.

## 5. Results and Discussion

This section describes the experiments performed to evaluate the performance of IDEA-I with the datasets published at (IEEE Dataport: https://ieee-dataport.org/documents/cyber-physical-dataset-mitm-attacks-power-systems, accessed on: 1 December 2021). The four use cases were implemented with MiTM attacks in the emulated synthetic electric grid. Different rules of combination and decision criteria are evaluated from the DSTE. Furthermore, these criteria or scores are used to compare different classifiers that must be considered prior to the incorporation of DS rules of combination. Post combination, the scores are fed to the NSGA-2 algorithm for feature selection to improve the performance. Based on the selected classifier, different types of fusion techniques used in DSTE are evaluated. The performance of two architectures introduced in the previous section that involve fusion by location and domain are assessed. Finally, the impact of the time resolution in the fusion operation on its accuracy is evaluated.

### 5.1. Decision Criteria Selection

In DSTE, different criteria or scores serve different purposes. Hence, four decision criteria: (a) Belief score, (b) Plausibility score, (c) Pignistic score, and (d) GBT score are evaluated. We evaluate the decision function for all the use cases, while considering different classifiers and cyber-physical features, and select the criteria that has the highest accuracy in the most scenarios. For the evaluations, the time resolution res is assumed to be 15 s, and the disjunctive rule of combination is considered. Table 1 shows that for seven classifiers (Support Vector Classifier (SVC), k-Nearest Neighbor (k-NN), Decision Tree (DT), Random Forest (RF), Gaussian Naive Bayes (GNB), Bernoulli Naive Bayes (BNB), and Multi-Layer Perceptron (MLP)) and 14 use cases (for varying polling interval, PI, in seconds and number of DNP3 outstations polled, os), the *pignistic score* has the highest accuracy under 78 scenarios. The *Belief score*, *Plausibility score*, and *GBT score* have the highest accuracy under seven, seven, and eight scenarios, respectively. Thus, results show the pignistic score as a reliable criteria for evaluation. Figure 4 shows the decision metrics for the disjunctive fusion with mass function computed from Decision Tree classifier.

### 5.2. Comparison by Classifier

A classifier’s performance may vary based on the features selected for training. Certain use cases perform well with the use of linear classifiers such as logistic regression, while some outperform with the use of non-linear classifiers such as DT or RF, and some with the use of the deep learning-based classifiers, depending on the spatial and temporal nature of features and relationship with the labels. Here, the seven classifiers are compared with the FL architecture of fusion. Figure 5a–d show the comparison of the classifiers used, prior to the disjunctive-based fusion, evaluated based on the precision, recall, F1-score, and accuracy. For all four use cases, the DT and RF classifiers are found to be better options with disjunctive-based fusion. However, the precision scores for UC-1 and UC-2 are better with SVM and k-NN classifiers, but accuracy and F1-score are considered the major criteria here, so DT and RF classifiers are preferred.

### 5.3. Comparison of Rules of Combination

Disjunctive rules of combination require at least one reliable piece of evidence. Conjunctive rules perform well if the evidence is independent and reliable. Cautious combination rules perform better when the cardinality of the frame of discernment is high. Hence, the comparison of conjunctive, disjunctive, and cautious conjunctive fusion techniques is performed. Figure 6a–c show the impact on the precision, recall, F1 score, and accuracy scores computed with the pignistic function for three different rules of combination. In the experiment, the intruder compromises the substation network; hence, both the substation router and the DNP3 outstation are compromised. Among the three sensors, with the assistance of one source considered to be secure, i.e., the DNP3 master, the disjunctive fusion with DT and RF classifiers performs better in comparison to conjunctive and its cautious counterpart.

### 5.4. Comparison of Two Architectures

Feature-based fusion is performed prior to fusion by location (FL). If the features from diverse domains are unable to be fused due to lack of performance or due to lack of evidence from any one domain, one needs to adopt the FLD architecture, where fusion-by-location on the raw domain-specific features are performed prior to fusion-by-domain. Figure 5a–d show the results for the FL architecture, while Figure 7a–d show results from the FLD architecture. FLD-based fusion outperforms FL-based fusion in many scenarios, but in some cases, there was not much influence. Hence, both may be adopted depending on the scenario.

### 5.5. Impact of Time Resolution while Merging by Location

Since low time resolution results in noise in intrusion detection, it is advisable to consider smoothening techniques. The physical sensor and cyber sensor time intervals between samples may vary; hence, it is essential for fusion to bring the samples to the same time frame. This comparison evaluates detection performance based on varying time resolution, which is considered during time synchronization prior to fusion by location. Figure 8a–d show the effect of different resolutions res in the *Mean Value-Based Time Synchronization* block, which is implemented for the probability scores obtained from the DT classifier for UC110OS30. Results show that increasing the sample time leads to better decision scores except for the GBT score.

### 5.6. Comparison with NSGA-2 Based Feature Selection

Detection performance considering feature selection using the NSGA-2 algorithm varies based on the selection of type of classifier. For all the classifiers, the results obtained with the GA algorithm improve the detection performance. The comparison of the results for RT and DF classifiers with and without GA-based feature selection is shown in Figure 9. In the classifier without GA-based feature selection, since all the features are selected, the amount of false negatives is relatively higher than the scenario when critical features are filtered out and considered using GA-based feature selection. For instance, in the DT, more features results in a bigger tree with more binary splits, making it difficult to obtain the optimal split that has the lowest cost. Hence this GA-based technique result in pruning or removing branches in the DT by increasing predictive power by reducing overfitting. Similar performances are observed for RF, which is a collection of DT. In the optimization problem, if the number of features are constrained using upper and lower bound, the performance will vary depending on the use-case scenario.

## 6. DSTE Evaluation Framework

A desktop application for IDEA-I is developed for the evaluation of DSTE rules of combination for different use-cases and parameters. Figure 10 shows the application, visualizing the decision metric for UC2_5OS_30poll for disjunctive fusion, with only cyber features, and using a mean value based time synchronization res of 15 s. The check-box labeled *“Merge by Location and Domain?"* is used to select either the FL- or FLD-based architecture. The code for the DSTE evaluation application has been made available in Github (Github: https://github.com/Abhijeet1990/DS_Fusion_GUI.git, accessed on 1 January 2022).

## 7. Conclusions

An evidence theoretic-based data fusion framework for detecting cyber intrusion in power systems is presented. The framework is evaluated by studying the performance of different classifiers using DS rules of combination. Results show the evidence from the DT and RF classifiers to be the best among other techniques. Results also show that higher time resolution in mean-value based time synchronization improves the decision metrics. The *pignistic function* decision criteria is observed to be the best among all the others for all the use cases. The FLD (autonomous architecture) outperforms the FL (centralized architecture)-based fusion in many scenarios, but in some, there is not much influence, so both techniques may be considered depending on the scenarios. Among the different rules of combination, the *disjunctive rules* performed the best when considered with *DT* and *RF* probability scores. Finally, an application has been developed and presented that performs these analyses and facilitates the DS theoretic framework for the fusion of cyber and physical sensors in power systems.

## Figures and Tables

**Figure 1 sensors-22-02100-f001:**
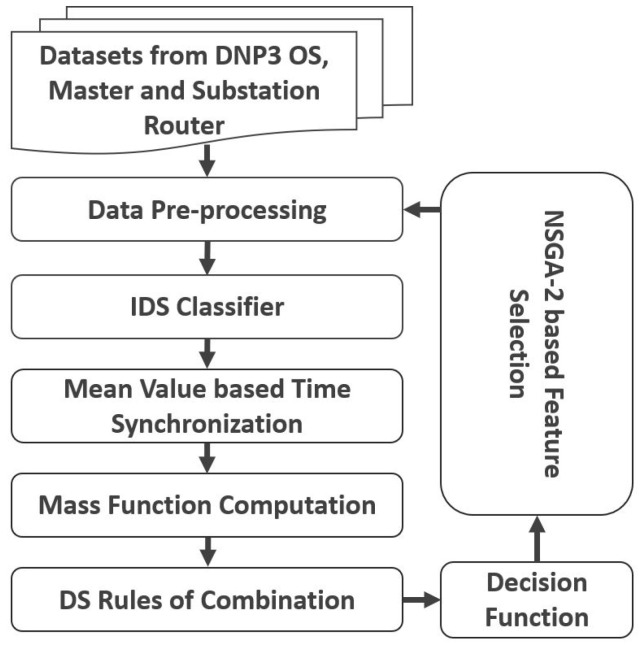
Sequence of operations.

**Figure 2 sensors-22-02100-f002:**
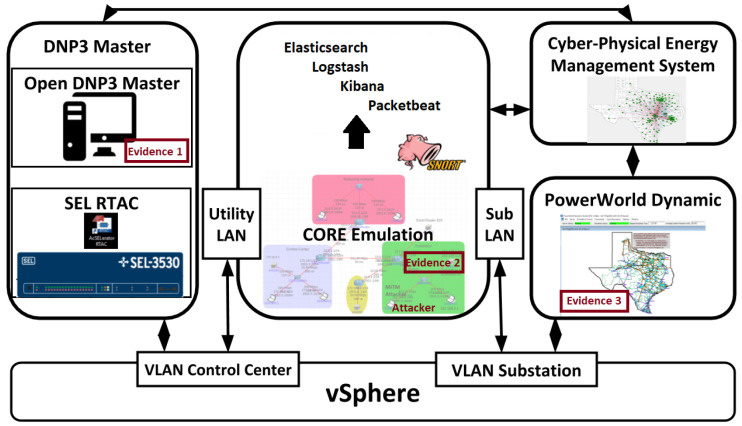
Testbed architecture showing three evidences: DNP3 Master (Evidence 1), Substation Router (Evidence 2), and PWDS emulating DNP3 outstations (Evidence 3).

**Figure 3 sensors-22-02100-f003:**
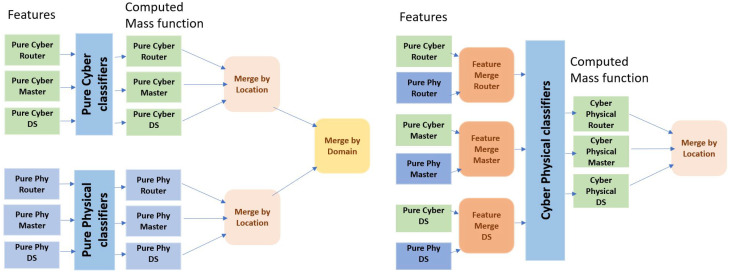
(**Left**) Fusion by location and domain (FLD) from pure cyber and physical classifiers. (**Right**) Fusion by location (FL) from cyber-physical classifiers.

**Figure 4 sensors-22-02100-f004:**
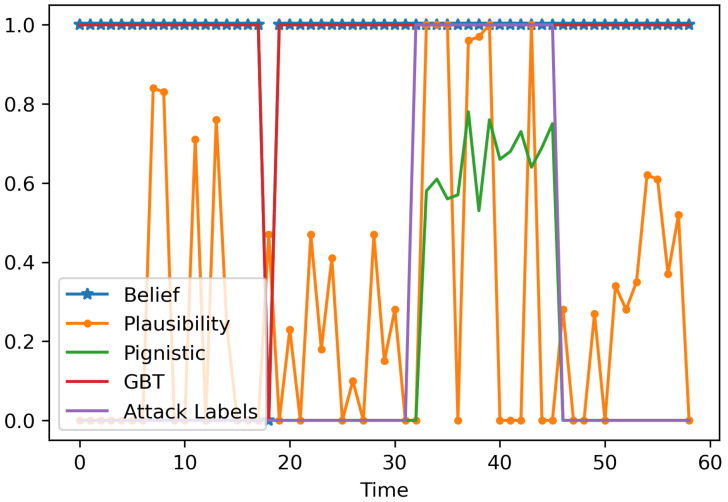
Decision criteria after distinctive fusion, with the probability scores from DT classifiers for Use Case 1.

**Figure 5 sensors-22-02100-f005:**
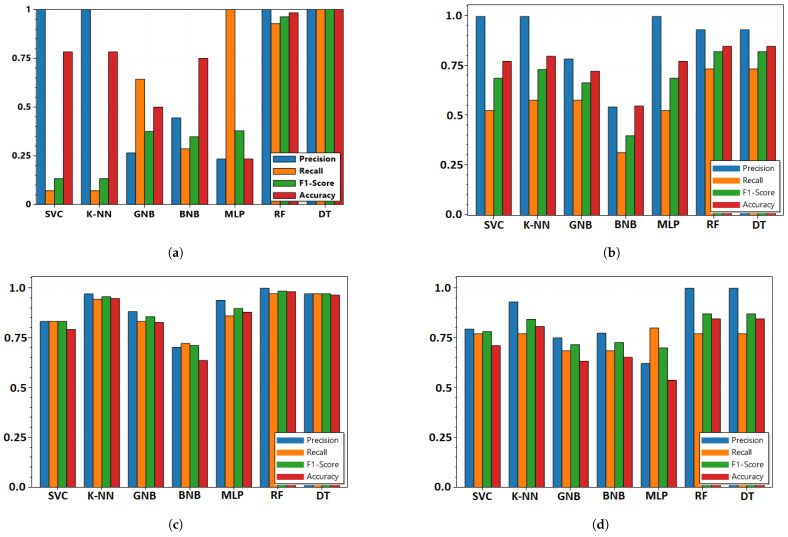
Precision, recall, F1 score, and accuracy obtained based on the pignistic score from disjunctive-based DS fusion performed from probability scores from different classifiers with cyber-physical features combined using FL architecture in Figure 3 for (**a**) UC1; (**b**) UC2; (**c**) UC3; and (**d**) UC4 with 30 s polling interval and 10 polled outstations.

**Figure 6 sensors-22-02100-f006:**
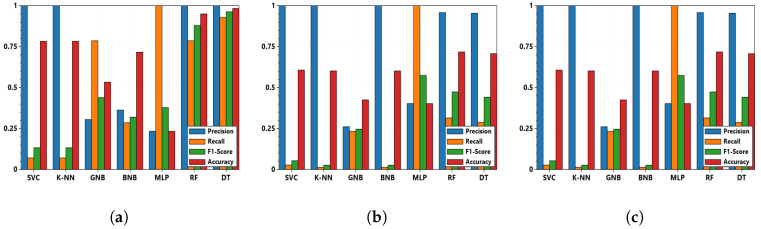
Precision, recall, F1 score, and accuracy obtained based on the pignistic score from three different rules of combination: (**a**) disjunctive; (**b**) conjunctive; (**c**) cautious conjunctive; tested with UC1, 30 s polling interval and 10 polled outstations.

**Figure 7 sensors-22-02100-f007:**
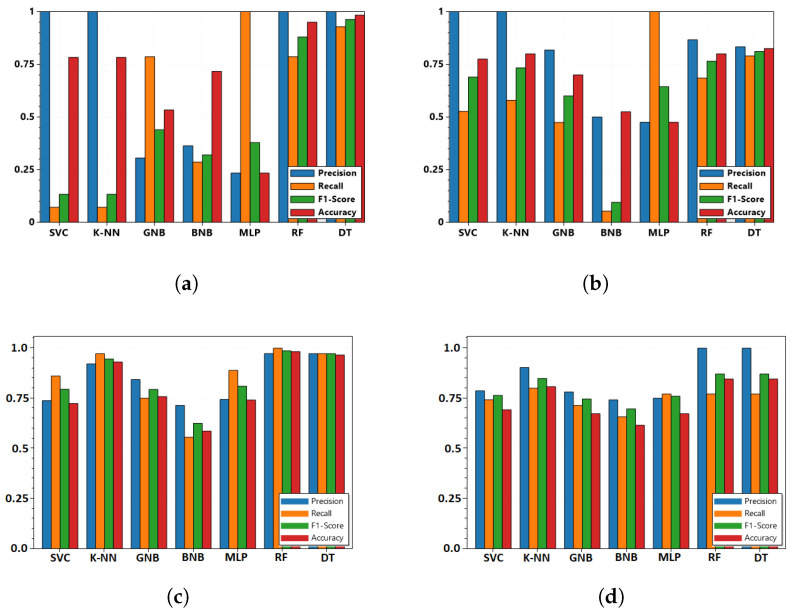
Precision, recall, F1 score, and accuracy obtained based on the pignistic score from disjunctive-based DS fusion by location and domain performed from probability scores from different classifiers with pure cyber and pure physical features combined using **FLD architecture** in Figure 3 for (**a**) UC1; (**b**) UC2; (**c**) UC3; (**d**) UC4.

**Figure 8 sensors-22-02100-f008:**
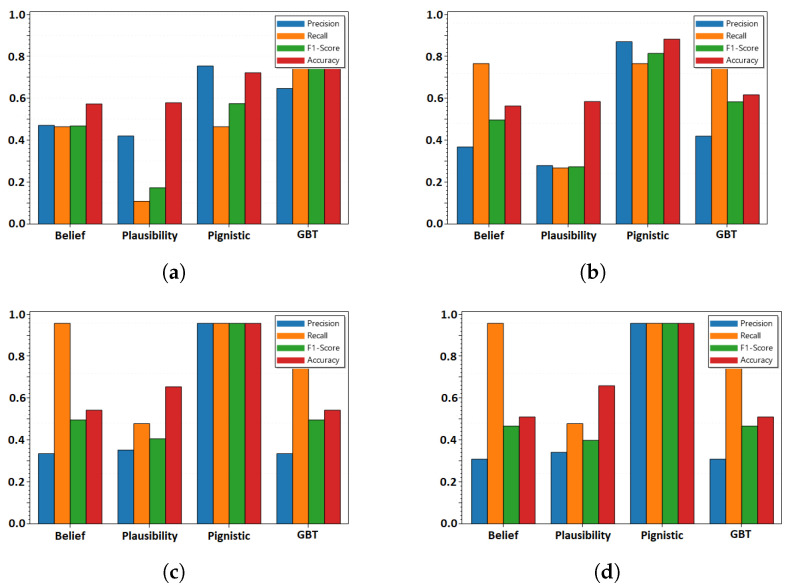
Precision, recall, F1 score, and accuracy obtained based on the different decision criteria from disjunctive-based DS fusion by location from probability scores of Decision Tree classifiers with combined cyber-physical features and different time resolutions (**a**) res = 5 s; (**b**) res = 10 s; (**c**) res = 15 s; (**d**) res = 20 s.

**Figure 9 sensors-22-02100-f009:**
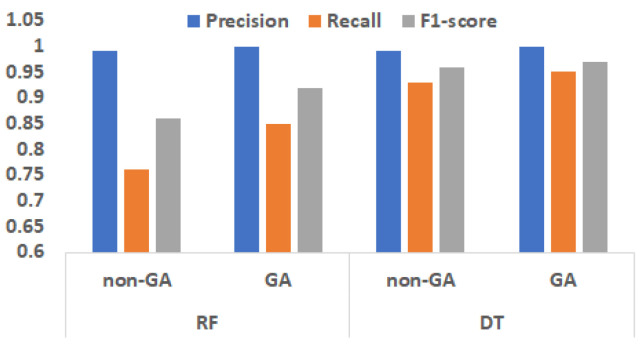
Comparison of the RF and DT-based classifier with and without the use of GA-based feature selection.

**Figure 10 sensors-22-02100-f010:**
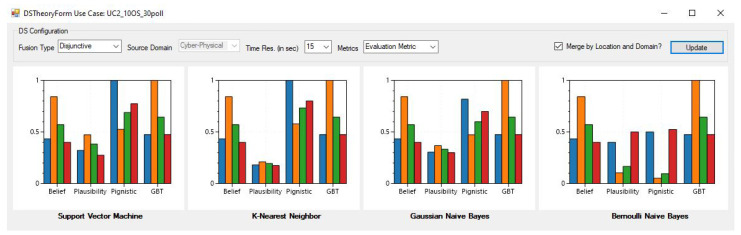
IDEA-I application for evaluating DSTE components for different scenarios, classifiers, and fusion architectures.

**Table 1 sensors-22-02100-t001:** Decision criteria with maximum accuracy evaluated accross different scenarios and classifiers (a = Belief score, b = Plausibility score, c = Pignistic score, d = GBT score). The detailed scores are available in the attached Appendix A.

**Scenarios**			**Classifiers**						
**UC**	**os**	**PI**	**SVC**	**K-NN**	**DT**	**RF**	**GNB**	**BNB**	**MLP**
UC1	10	30	c	c	c	c	b	c	b
	10	60	c	c	c	c	c	c	c
UC2	5	30	c	c	c	c	c	c	c
	5	60	c	c	d	c	c	c	c
	10	30	c	c	a	c	c	c	c
	10	60	c	c	a	c	b	c	c
UC3	5	30	c	c	c	c	c	c	c
	5	60	c	c	a	c	c	c	c
	10	30	b	c	c	c	c	a	c
	10	60	b	c	c	c	c	b	c
UC4	5	30	c	c	a	c	d	b	d
	5	60	c	c	a	c	c	c	d
	10	30	c	c	d	c	d	d	d
	10	60	c	c	c	c	c	c	a

## Data Availability

The dataset for the four use cases with the Man-in-The-Middle attack emulation in a synthetic electric grid are available at IEEE Dataport [34], while the code for the application that presents the DSTE framework is available at Github [35].

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
