# Peer review of "Inter-Domain Fusion for Enhanced Intrusion Detection in Power Systems: An Evidence Theoretic and Meta-Heuristic Approach"

_sensors, 2022, doi:10.3390/s22062100_

Round 1

Reviewer 1 Report

Dear authors:

Thanks for your work. It is an interesting well written manuscript. I have small comments which I hope you can fix it. 

1- Table 1, for the sake of consistency, UC1 missing values compared to the rest of UCs.

2

Figure 5 a) MLP missing: Precision, Recall, and F1 score

Figure 6 a) MLP missing: Precision, Recall, and F1 score

  1.  b)  MLP missing: Precision, Recall, and F1 score
  2.  c) K-NN and MLP missing: Precision, Recall, and F1 score

 Figure 7 a) MLP missing: Precision, Recall, and F1 score

  1. b) MLP missing: Precision, Recall, and F1 score

Thank you and good luck!

Author Response

Dear Reviewer,

I have attached the response as an attachment.

Regards,

Abhijeet Sahu

Reviewer 2 Report

Dear Author,
With appropriate datasets, experiments, and references, the author has proposed Inter-Domain Fusion for Enhanced Intrusion Detection.
As a result, I recommend that you accept with minor revision. 

Comments: 
"Figure 9. Comparison of the RF and DT based classifier with and without the use of GA-based feature
selection." This requires atleast one paragraph of explanation.

"Table 1. Decision criteria with maximum accuracy evaluated accross different scenarios and classifiers
(a = Belief score, b= Plausibility score, c= Pignistic score; d= GBT score)." Here, the score values of a, b, c, d may be supplied in separate table and this four new table can be placed in the supplementary files. This allows the reader to comprehend better.

Reviewer 3 Report

The authors focus their study on introducing an interdomain evidence theoretic approach for inference which reframes the detection problem in order to make good decisions considering the uncertainty that exists in intrusion detection systems. The proposed approach combines the Dempster Shafer combination rules and their variants in order to reduce the false alerts.

The manuscript is overall well written and easy to follow and the authors have well thought out their main contributions. The provided analysis of the proposed solution is concrete, complete, and correct. The provided numerical results in order to show the drawbacks of benefits of the proposed framework are rich and convincing.

The authors should consider the following suggestions provided by the reviewer in order to improve the scientific depth of their manuscript, as well as they should address the following comments in order to improve the quality of presentation of their manuscript.

Initially, the authors introduce a software based solution in order to reduce the false alerts. However, the concept of physical unclonable functions have has been introduced in the literature in order to support the authentication of devices, such as Artificially Intelligent Electronic Money, doi: 10.1109/MCE.2020.3024512, and has already been applied in several fields like e-cash protocols. Similarly, in this paper, the authors can adopt the physical unclonable functions in order to reduce the false alerts.

Based on the previous comment, the authors should improve the provided related work in order to demonstrate and discuss the software based and hardware based authentication solutions that exist in the literature.

Based on the previous comment, the authors should clarify the reasons that they base their proposed approach in a software based solution.

In section 3, where the authors should include an additional subsection providing the information and control overhead that is imposed to the system in order for the proposed framework to be implemented.

Based on the previous comment, the authors should quantify the information and control overhead in the section of the numerical results in order to demonstrate the computing overhead that is imposed to the system.

Finally, the overall manuscript should be checked for typos, syntax, and grammar errors in order to improve the quality of its presentation.

Round 2

Reviewer 3 Report

The authors have addressed the reviewers' comments in detail. The manuscript can be accepted.